# *Muffliato*: Peer-to-Peer Privacy Amplification for Decentralized Optimization and Averaging

**Edwige Cyffers [1],⋆ Mathieu Even[2],⋆, Aurélien Bellet [1], Laurent Massoulié [2]**

⋆ Equal contribution

[1]Univ. Lille, Inria, CNRS, Centrale Lille, UMR 9189 - CRIStAL, F-59000 Lille

[2]Inria Paris - Département d'informatique de l'ENS, PSL Research University, Paris, France

## Abstract

Decentralized optimization is increasingly popular in machine learning for its scalability and efficiency. Intuitively, it should also provide better privacy guarantees, as nodes only observe the messages sent by their neighbors in the network graph. But formalizing and quantifying this gain is challenging: existing results are typically limited to Local Differential Privacy (LDP) guarantees that overlook the advantages of decentralization. In this work, we introduce pairwise network differential privacy, a relaxation of LDP that captures the fact that the privacy leakage from a node $u$ to a node $v$ may depend on their relative position in the graph. We then analyze the combination of local noise injection with (simple or randomized) gossip averaging protocols on fixed and random communication graphs. We also derive a differentially private decentralized optimization algorithm that alternates between local gradient descent steps and gossip averaging. Our results show that our algorithms amplify privacy guarantees as a function of the distance between nodes in the graph, matching the privacy-utility trade-off of the trusted curator, up to factors that explicitly depend on the graph topology. Remarkably, these factors become constant for expander graphs. Finally, we illustrate our privacy gains with experiments on synthetic and real-world datasets.

## 1 Introduction

Training machine learning models traditionally requires centralizing data in a single server, raising issues of scalability and privacy. An alternative is to use Federated Learning (FL), where each user keeps her data on device [44, 36]. In *fully decentralized* FL, the common hypothesis of a central server is also removed, letting users, represented as nodes in a graph, train the model via peer-to-peer communications along edges. This approach improves scalability and robustness to central server failures, enabling lower latency, less power consumption and quicker deployment [43, 11, 54, 50, 1, 42, 39].

Another important dimension is privacy, as a wide range of applications deal with sensitive and personal data. The gold standard to quantify the privacy leakage of algorithms is Differential Privacy (DP) [20]. DP typically requires to randomly perturb data-dependent computations to prevent the final model from leaking too much information about any individual data point (e.g., through data memorization). However, decentralized algorithms do not only reveal the final model to the participating nodes, but also the results of some intermediate computations. A standard solution is to use Local Differential Privacy (LDP) [37, 19], where random perturbations are performed locally by each user, thus protecting against an attacker that would observe everything that users share. From the

36th Conference on Neural Information Processing Systems (NeurIPS 2022).

algorithmic standpoint, local noise addition can be easily performed within decentralized algorithms, as done for instance in [34, 6, 15, 60, 58]. Unfortunately, LDP requires large amounts of noise, and thus provides poor utility.

In this work, we show that LDP guarantees give a very pessimistic view of the privacy offered by decentralized algorithms. Indeed, there is no central server receiving all messages, and the participating nodes can only observe the messages sent by their neighbors in the graph. So, a given node should intuitively leak less information about its private data to nodes that are far away. We formally quantify this privacy amplification for the fundamental brick of communication at the core of decentralized optimization: gossip algorithms. Calling *Muffliato* the combination of local noise injection with a gossip averaging protocol, we precisely track the resulting privacy leakage between each pair of nodes. Through gossiping, the private values and noise terms of various users add up, obfuscating their contribution well beyond baseline LDP guarantees: as their distance in the graph increases, the privacy loss decreases. We then show that the choice of graph is crucial to enforce a good privacy-utility trade-off while preserving the scalability of gossip algorithms.

Our results are particularly attractive in situations where nodes want stronger guarantees with respect to some (distant) peers. For instance, in social network graphs, users may have lower privacy expectations with respect to close relatives than regarding strangers. In healthcare, a patient might trust her family doctor more than she trusts other doctors, and in turn more than employees of a regional agency and so on, creating a hierarchical level of trust that our algorithms naturally match.

## 1.1 Contributions and outline of the paper

*(i)* Inspired by the recent definitions of Cyffers and Bellet [17], we introduce *pairwise network DP*, a relaxation of Local Differential Privacy which is able to quantify the privacy loss of a decentralized algorithm for each pair of distinct users in a graph.

*(ii)* We propose *Muffliato*[1], a privacy amplification mechanism composed of local Gaussian noise injection at the node level followed by gossiping for averaging the private values. It offers privacy amplification that increases as the distance between two nodes increases. Informally, the locally differentially private value shared by a node $u$ is mixed with other contributions, to the point that the information that leaks to another node $v$ can have a very small sensitivity to the initial value in comparison to the accumulated noise.

*(iii)* We analyze both synchronous gossip [18] and randomized gossip [11] under a unified privacy analysis with arbitrary time-varying gossip matrices. We show that the magnitude of the privacy amplification is significant: the average privacy loss over all the pairs in this setting reaches the optimal privacy-utility trade-off of a trusted aggregator, up to a factor $\frac{d}{\sqrt{\lambda_W}}$, where $\lambda_W$ is the weighted graph eigengap and $d$ the maximum degree of the graph. Remarkably, this factor can be of order 1 for expanders, yielding a sweet spot in the privacy-utility-scalability trade-off of gossip algorithms. Then, we study the case where the graph is itself random and private, and derive stronger privacy guarantees.

*(iv)* Finally, we develop and analyze differentially private decentralized Gradient Descent (GD) and Stochastic Gradient Descent (SGD) algorithms to minimize a sum of local objective functions. Building on *Muffliato*, our algorithms alternate between rounds of differentially private gossip communications and local gradient steps. We prove that they enjoy the same privacy amplification described above for averaging, up to factors that depend on the regularity of the global objective.

*(v)* We demonstrate the usefulness of our approach and analysis through experiments on synthetic and real-world datasets and network graphs. We illustrate how privacy is amplified between nodes in the graph as a function of their distance, and show how time-varying random graphs can be used to split the privacy loss more uniformly across nodes in decentralized optimization.

## 1.2 Related work

*Gossip algorithms and decentralized optimization.* Gossip algorithms [10, 18] were introduced to compute the global average of local vectors through peer-to-peer communication, and are at the core of many decentralized optimization algorithms. Classical decentralized optimization algorithms alternate

---

[1]The name is borrowed from the Harry Potter series: it designates a "spell that filled the ears of anyone nearby with an unidentifiable buzzing", thereby concealing messages from unintended listeners through noise injection.

between gossip communications and local gradient steps [48, 38, 39], or use dual formulations and formulate the consensus constraint using gossip matrices to obtain decentralized dual or primal-dual algorithms [54, 32, 24, 25, 40, 1]. We refer the reader to [49] for a broader survey on decentralized optimization. Our algorithms are based on the general analysis of decentralized SGD in [39].

*LDP and privacy amplification mechanisms.* Limitations of LDP for computing the average of the private values of $n$ users have been studied, showing that for a fixed privacy budget, the expected squared error in LDP is $n$ times larger than in central DP [12]. More generally, LDP is also known to significantly reduce utility for many learning problems [61, 57], which motivates the study of interme-diate trust models. Cryptographic primitives, such as secure aggregation [21, 55, 9, 13, 35, 5, 53] and secure shuffling [16, 23, 3, 31, 30], as well as additional mechanisms such as amplification by subsam-pling [2] or amplification by iteration [28], can offer better utility for some applications, but cannot be easily applied in a fully decentralized setting, as they require coordination by a central server.

*Privacy amplification through decentralization.* The idea that decentralized communications can provide differential privacy guarantees was initiated by [7] in the context of rumor spreading. Closer to our work, [17] showed privacy amplification for random walk algorithms on complete graphs, where the model is transmitted from one node to another sequentially. While we build on their notion of Network DP, our work differs from [17] in several aspects: (i) our analysis holds for any graph and explicitly quantifies its effect, (ii) instead of worst-case privacy across all pairs of nodes, we prove pairwise guarantees that are stronger for nodes that are far away from each other, and (iii) unlike random walk approaches, gossip algorithms allow parallel computation and thus better scalability.

## 2  Setting and Pairwise Network Differential Privacy

We study a decentralized model where $n$ nodes (users) hold private datasets and communicate through gossip protocols, that we describe in Section 2.1. In Section 2.2, we recall differential privacy notions and the two natural baselines for our work, central and local DP. Finally, we introduce in Section 2.3 the relaxation of local DP used throughout the paper: the *pairwise network DP*.

### 2.1  Gossip Algorithms

We consider a connected graph $G = (\mathcal{V}, \mathcal{E})$ on a set $\mathcal{V}$ of $n$ users. An edge $\{u, v\} \in \mathcal{E}$ indicates that $u$ and $v$ can communicate (we say they are neighbors). Each user $v \in \mathcal{V}$ holds a local dataset $\mathcal{D}_v$ and we aim at computing averages of private values. This averaging step is a key building block for solving machine learning problems in a decentralized manner, as will be discussed in Section 4. From any graph, we can derive a gossip matrix.

**Definition 1** (Gossip matrix). *A gossip matrix over a graph $G$ is a* symmetric *matrix $W \in \mathbb{R}^{n \times n}$ with non-negative entries, that satisfies $W\mathbb{1} = \mathbb{1}$ i.e. $W$ is stochastic ($\mathbb{1} \in \mathbb{R}^n$ is the vector with all entries equal to 1), and such that for any $u, v \in \mathcal{V}$, $W_{u,v} > 0$ implies that $\{u, v\} \in \mathcal{E}$ or $u = v$.*

The iterates of synchronous gossip [18] are generated through a recursion of the form $x^{t+1} = Wx^t$, and converge to the mean of initial values $x^0 \in \mathbb{R}^n$ at a linear rate $e^{-t\lambda_W}$, with $\lambda_W$ defined below.

**Definition 2** (Spectral gap). *The spectral gap $\lambda_W$ associated with a gossip matrix $W$ is $\min_{\lambda \in \mathrm{Sp}(W) \setminus \{1\}} (1 - |\lambda|)$, where $\mathrm{Sp}(W)$ is the spectrum of $W$.*

The inverse of $\lambda_W$ is the relaxation time of the random walk on $G$ with transition probabilities $W$, and is closely related to the connectivity of the graph: adding edges improve mixing properties ($\lambda_W$ increases), but can reduce scalability by increasing node degrees (and thus the per-iteration communication complexity). The rate of convergence can be accelerated to $e^{-t\sqrt{\lambda_W}}$ using re-scaled Chebyshev polynomials, leading to iterates of the form $x^t = P_t(W)x^0$ [8].

**Definition 3** (Re-scaled Chebyshev polynomials). *The re-scaled Chebyshev polynomials $(P_t)_{t \geqslant 0}$ with scale parameter $\gamma \in [1, 2]$ are defined by second-order linear recursion:*

$$P_0(X) = 1, \quad P_1(X) = X, \quad P_{t+1}(X) = \gamma X P_t(X) + (1 - \gamma) P_{t-1}(X), \, t \geqslant 2. \quad (1)$$

### 2.2  Rényi Differential Privacy

Differential Privacy (DP) quantifies how much information the output of an algorithm $\mathcal{A}$ leaks about the dataset taken as input [20]. DP requires to define an adjacency relation between datasets. In this

work, we adopt a user-level relation [45] which aims to protect the whole dataset $\mathcal{D}_v$ of a given user represented by a node $v \in \mathcal{V}$. Formally, $\mathcal{D} = \cup_{v \in \mathcal{V}} \mathcal{D}_v$ and $\mathcal{D}' = \cup_{v \in \mathcal{V}} \mathcal{D}'_v$ are adjacent datasets, denoted by $\mathcal{D} \sim \mathcal{D}'$, if there exists $v \in \mathcal{V}$ such that only $\mathcal{D}_v$ and $\mathcal{D}'_v$ differ. We use $\mathcal{D} \sim_v \mathcal{D}'$ to denote that $\mathcal{D}$ and $\mathcal{D}'$ differ only in the data of user $v$.

We use Rényi Differential Privacy (RDP) [46] to measure the privacy loss, which allows better and simpler composition than the classical $(\epsilon, \delta)$-DP. Note that any $(\alpha, \varepsilon)$-RDP algorithm is also $(\varepsilon + \ln(1/\delta)/(\alpha - 1), \delta)$-DP for any $0 < \delta < 1$ [46].

**Definition 4** (Rényi Differential Privacy). *An algorithm $\mathcal{A}$ satisfies $(\alpha, \varepsilon)$-Rényi Differential Privacy (RDP) for $\alpha > 1$ and $\varepsilon > 0$ if for all pairs of neighboring datasets $\mathcal{D} \sim \mathcal{D}'$:*

$$D_\alpha\left(\mathcal{A}(\mathcal{D}) \| \mathcal{A}(\mathcal{D}')\right) \leqslant \varepsilon \,, \tag{2}$$

*where for two random variables $X$ and $Y$, $D_\alpha(X \| Y)$ is the Rényi divergence between $X$ and $Y$:*

$$D_\alpha(X \| Y) = \tfrac{1}{\alpha-1} \ln \int \left(\tfrac{\mu_X(z)}{\mu_Y(z)}\right)^\alpha \mu_Y(z) dz \,.$$

*with $\mu_X$ and $\mu_Y$ the respective densities of $X$ and $Y$.*

Without loss of generality, we consider gossip algorithms with a single real value per node (in that case, $\mathcal{D}_v = \{x_v\}$ for some $x_v \in \mathbb{R}$), and we aim at computing a private estimation of the mean $\bar{x} = (1/n) \sum_v x_v$. The generalization to vectors is straightforward, as done subsequently for optimization in Section 4. In general, the value of a (scalar) function $g$ of the data can be privately released using the Gaussian mechanism [20, 46], which adds $\eta \sim \mathcal{N}(0, \sigma^2)$ to $g(\mathcal{D})$. It satisfies $(\alpha, \alpha\Delta_g^2/2\sigma^2)$-RDP for any $\alpha > 1$, where $\Delta_g = \sup_{\mathcal{D} \sim \mathcal{D}'} \|g(\mathcal{D}) - g(\mathcal{D}')\|$ is the sensitivity of $g$. We focus on the Gaussian mechanism for its simplicity (similar results could be derived for other DP mechanisms), and thus assume an upper bound on the $L_2$ inputs sensitivity.

**Assumption 1.** *There exists some constant $\Delta > 0$ such that for all $u \in \mathcal{V}$ and for any adjacent datasets $\mathcal{D} \sim_u \mathcal{D}'$, we have $\|x_u - x'_u\| \leqslant \Delta$.*

In central DP, a trusted aggregator can first compute the mean $\bar{x}$ (which has sensitivity $\Delta/n$) and then reveal a noisy version with the Gaussian mechanism. On the contrary, in local DP where there is no trusted aggregator and everything that a given node reveals can be observed, each node must locally perturb its input (which has sensitivity $\Delta$), deteriorating the privacy-utility trade-off. Formally, to achieve $(\alpha, \varepsilon)$-DP, one cannot have better utility than:

$$\mathbb{E}\left[\left\|x^{\text{out}} - \bar{x}\right\|^2\right] \leqslant \frac{\alpha\Delta^2}{2n\varepsilon} \quad \text{for local DP}, \quad \text{and} \quad \mathbb{E}\left[\left\|x^{\text{out}} - \bar{x}\right\|^2\right] \leqslant \frac{\alpha\Delta^2}{2n^2\varepsilon} \quad \text{for central DP},$$

where $x^{\text{out}}$ is the output of the algorithm. This $1/n$ gap motivates the study of relaxations of local DP.

### 2.3 Pairwise Network Differential Privacy

We relax local DP to take into account privacy amplification between nodes that are distant from each other in the graph. We define a decentralized algorithm $\mathcal{A}$ as a randomized mapping that takes as input a dataset $\mathcal{D} = \cup_{v \in \mathcal{V}}(\mathcal{D}_v)$ and outputs the transcript of all messages exchanged between users in the network. A message between neighboring users $\{u, v\} \in \mathcal{E}$ at time $t$ is characterized by the tuple $(u, m(t), v)$: user $u$ sent a message with content $m(t)$ to user $v$, and $\mathcal{A}(\mathcal{D})$ is the set of all these messages. Each node $v$ only has a partial knowledge of $\mathcal{A}(\mathcal{D})$, captured by its *view*:

$$\mathcal{O}_v\left(\mathcal{A}(\mathcal{D})\right) = \{(u, m(t), v) \in \mathcal{A}(\mathcal{D}) \quad \text{such that} \quad \{u, v\} \in \mathcal{E}\} \,.$$

This subset corresponds to direct interactions of $v$ with its neighbors, which provide only an indirect information on computations in others parts of the graph. Thus, we seek to express privacy constraints that are personalized for each pair of nodes. This is captured by our notion of Pairwise Network DP.

**Definition 5** (Pairwise Network DP). *For $f : \mathcal{V} \times \mathcal{V} \to \mathbb{R}^+$, an algorithm $\mathcal{A}$ satisfies $(\alpha, f)$-Pairwise Network DP (PNDP) if for all pairs of distinct users $u, v \in \mathcal{V}$ and neighboring datasets $D \sim_u D'$:*

$$D_\alpha\left(\mathcal{O}_v(\mathcal{A}(\mathcal{D})) \| \mathcal{O}_v(\mathcal{A}(\mathcal{D}'))\right) \leqslant f(u, v) \,. \tag{3}$$

*We note $\varepsilon_{u \to v} = f(u, v)$ the privacy leaked to $v$ from $u$ and say that $u$ is $(\alpha, \varepsilon_{u \to v})$-PNDP with respect to $v$ if only inequality (3) holds for $f(u, v) = \varepsilon_{u \to v}$.*

By taking $f$ constant in Definition 5, we recover the definition of Network DP [17]. Our pairwise variant refines Network DP by allowing the privacy guarantee to depend on $u$ and $v$ (typically, on their relative position in the graph). We assume that users are *honest but curious*: they truthfully follow the protocol, but may try to derive as much information as possible from what they observe. We refer to Appendix G for a natural adaptation of our definition and results to the presence of *colluding nodes* and to the protection of *groups* of users.

In addition to pairwise guarantees, we will use the *mean privacy loss* $\overline{\varepsilon}_v = \frac{1}{n} \sum_{u \in \mathcal{V} \setminus \{v\}} f(u,v)$ to compare with baselines LDP and trusted aggregator by enforcing $\overline{\varepsilon} = \max_{v \in \mathcal{V}} \overline{\varepsilon}_v \leqslant \varepsilon$. The value $\overline{\varepsilon}_v$ is the average of the privacy loss from all the nodes to $v$ and thus does not correspond to a proper privacy guarantee, but it provides a convenient way to summarize our gains, noting that distant nodes — in ways that will be specified — will have better privacy guarantee than this average, while worst cases will remain bounded by the baseline LDP guarantee provided by local noise injection.

## 3 Private Gossip Averaging

In this section, we analyze a generic algorithm with arbitrary time-varying communication matrices for averaging. Then, we instantiate and discuss these results for synchronous communications with a fixed gossip matrix, communications using randomized gossip [11], and with Erdös-Rényi graphs.

### 3.1 General Privacy Analysis of Gossip Averaging

We consider gossip over time-varying graphs $(G_t)_{0 \leqslant t \leqslant T}$, defined as $G_t = (\mathcal{V}, \mathcal{E}_t)$, with corresponding gossip matrices $(W_t)_{0 \leqslant t \leqslant T}$. The *generic Muffliato* algorithm $\mathcal{A}^T$ for averaging $x = (x_v)_{v \in \mathcal{V}}$ corresponds to an initial noise addition followed by $T$ gossip steps. Writing $W_{0:t} = W_{t-1} \ldots W_0$, the iterates of $\mathcal{A}^T$ are thus defined by:

$$\forall v \in \mathcal{V}, x_v^0 = x_v + \eta_v \text{ with } \eta_v \sim \mathcal{N}(0, \sigma^2), \quad \text{and } x^{t+1} = W_t x^t = W_{0:t+1}(x + \eta). \quad (4)$$

Note that the update rule at node $v \in \mathcal{V}$ writes as $x_v^{t+1} = \sum_{w \in \mathcal{N}_t(v)} (W_t)_{v,w} x_w^t$ where $\mathcal{N}_t(v)$ are the neighbors of $v$ in $G_t$, so for the privacy analysis, the view of a node is:

$$\mathcal{O}_v\big(\mathcal{A}^T(\mathcal{D})\big) = \big\{ \big(W_{0:t}(x + \eta)\big)_w \mid \{v, w\} \in \mathcal{E}_t, \quad 0 \leqslant t \leqslant T - 1 \big\} \cup \{x_v\}. \quad (5)$$

**Theorem 1.** *Let $T \geqslant 1$ and denote by $\mathcal{P}_{\{v,w\}}^T = \{s < T : \{v, w\} \in \mathcal{E}_s\}$ the set of time-steps with communication along edge $\{v, w\}$. Under Assumption 1, $\mathcal{A}^T$ is $(\alpha, f)$-PNDP with:*

$$f(u,v) = \frac{\alpha \Delta^2}{2\sigma^2} \sum_{w \in \mathcal{V}} \sum_{t \in \mathcal{P}_{\{v,w\}}^T} \frac{(W_{0:t})_{u,w}^2}{\|(W_{0:t})_w\|^2}. \quad (6)$$

This theorem, proved in Appendix B, gives a tight computation of the privacy loss between every pair of nodes and can easily be computed numerically (see Section 5). Since distant nodes correspond to small entries in $W_{0:t}$, Equation 6 suggests that they reveal less to each other. We will characterize this precisely for the case of fixed communication graph in the next subsection.

Another way to interpret the result of Theorem 1 is to derive the corresponding mean privacy loss:

$$\overline{\varepsilon}_v = \frac{\alpha \Delta^2 T_v}{2n\sigma^2}, \quad (7)$$

where $T_v$ is the total number of communications node $v$ was involved with up to time $T$. Thus, in comparison with LDP, the mean privacy towards $v$ is $n/T_v$ times smaller. In other words, a node learns much less than in LDP as long as it communicates $o(n)$ times.

### 3.2 Private Synchronous *Muffliato*

We now consider *Muffliato* over a fixed graph (Algorithm 1). Note that we use gossip acceleration (see Definition 3). We start by analyzing the utility of *Muffliato*, which decomposes as an averaging error term vanishing exponentially fast, and a *bias* term due to the noise. General convergence rates are given in Appendix C, from which we extract the following result.

| **Algorithm 1:** MUFFLIATO | **Algorithm 2:** RANDOMIZED MUFFLIATO |
|---|---|

**Input:** local values $(x_v)_{v\in\mathcal{V}}$ to average, gossip matrix $W$ on a graph $G$, in $T$ iterations, noise variance $\sigma^2$

$\gamma \leftarrow 2\frac{1-\sqrt{\lambda_W(1-\frac{\lambda_W}{4})}}{(1-\lambda_W/2)^2}$

**for** *all nodes $v$ in parallel* **do**
  $\quad x_v^0 \leftarrow x_v + \eta_v$ where $\eta_v \sim \mathcal{N}(0,\sigma^2)$
**for** $t = 0$ *to* $T-1$ **do**
  **for** *all nodes $v$ in parallel* **do**
    **for** *all neighbors $w$ defined by $W$* **do**
      $\quad$ Send $x_v^t$, receive $x_w^t$
    $x_v^{t+1} \leftarrow$
      $(1-\gamma)x_v^{t-1} + \gamma\sum_{w\in\mathcal{N}_v}W_{v,w}x_w^t$

**Input:** local values $(x_v)_{v\in\mathcal{V}}$ to average, activation intensities $(p_{\{v,w\}})_{\{v,w\}\in\mathcal{E}}$, in $T$ iterations, noise variance $\sigma^2$

**for** *all nodes $v$ in parallel* **do**
  $\quad x_v^0 \leftarrow x_v + \eta_v$ where $\eta_v \sim \mathcal{N}(0,\sigma^2)$
**for** $t = 0$ *to* $T-1$ **do**
  Sample $\{v_t,w_t\}\in\mathcal{E}$ with probability $p_{\{v_t,w_t\}}$
  $v_t$ and $w_t$ exchange $x_{v_t}^t$ and $x_{w_t}^t$
  Local averaging:
    $x_{v_t}^{t+1} = x_{w_t}^{t+1} = \frac{x_{v_t}^{t+1}+x_{w_t}^{t+1}}{2}$
  For $v\in\mathcal{V}\setminus\{v_t,w_t\}$, $x_v^{t+1}=x_v^t$

Table 1: Utility of *Muffliato* for several topologies under the constraint $\bar{\varepsilon}\leqslant\varepsilon$ for the classic gossip matrix where $W_{v,w}=\min(1/d_v,1/d_w)$ and $d_v$ is the degree of node $v$. $\tilde{\mathcal{O}}(\cdot)$ hides constant and logarithmic factors. Recall that utility is $\tilde{\mathcal{O}}(\alpha\Delta^2/n\varepsilon)$ for LDP and $\tilde{\mathcal{O}}(\alpha\Delta^2/n^2\varepsilon)$ for central DP.

| **Graph** | Arbitrary | Expander | $C$-Torus | Complete | Ring |
|---|---|---|---|---|---|
| Algorithm 1 | $\tilde{\mathcal{O}}\left(\frac{\alpha\Delta^2 d}{n^2\varepsilon\sqrt{\lambda_W}}\right)$ | $\tilde{\mathcal{O}}\left(\frac{\alpha\Delta^2}{n^2\varepsilon}\right)$ | $\tilde{\mathcal{O}}\left(\frac{\alpha\Delta^2 C}{n^{2-1/C}\varepsilon}\right)$ | $\tilde{\mathcal{O}}\left(\frac{\alpha\Delta^2}{n\varepsilon}\right)$ | $\tilde{\mathcal{O}}\left(\frac{\alpha\Delta^2}{n\varepsilon}\right)$ |
| Algorithm 2 | $\tilde{\mathcal{O}}\left(\frac{\alpha\Delta^2}{n^2\varepsilon\lambda_W}\right)$ | $\tilde{\mathcal{O}}\left(\frac{\alpha\Delta^2}{n^2\varepsilon}\right)$ | $\tilde{\mathcal{O}}\left(\frac{\alpha\Delta^2}{n^{2-2/C}\varepsilon}\right)$ | $\tilde{\mathcal{O}}\left(\frac{\alpha\Delta^2}{n^2\varepsilon}\right)$ | $\tilde{\mathcal{O}}\left(\frac{\alpha\Delta^2}{n\varepsilon}\right)$ |

**Theorem 2** (Utility analysis). *Let $\lambda_W$ be the spectral gap of $W$. Muffliato (Algorithm 1) verifies, for any $t\geqslant T^{\mathrm{stop}}$:*

$$\frac{1}{2n}\sum_{v\in\mathcal{V}}\mathbb{E}\left[\left\|x_v^t-\bar{x}\right\|^2\right]\leqslant\frac{3\sigma^2}{n}\,,\quad\text{where}\quad T^{\mathrm{stop}}=\frac{1}{\sqrt{\lambda_W}}\ln\left(\frac{n}{\sigma^2}\max\left(\sigma^2,\frac{1}{n}\sum_{v\in\mathcal{V}}\|x_v-\bar{x}\|^2\right)\right).$$

For the privacy guarantees, Theorem 1 still holds as accelerated gossip can be seen as a post-processing of the non-accelerated version. Thanks to the fixed graph, we can derive a more explicit formula.

**Corollary 1.** *Algorithm 1 satisfies $(\alpha,\varepsilon_{u\to v}^T(\alpha))$-PNDP for node $u$ with respect to $v$, with:*

$$\varepsilon_{u\to v}^T(\alpha)\leqslant\frac{\alpha\Delta^2 n}{2\sigma^2}\max_{\{v,w\}\in\mathcal{E}}W_{v,w}^{-2}\sum_{t=1}^{T}\mathbb{P}\left(X^t=v|X^0=u\right)^2\,,$$

*where $(X^t)_t$ is the random walk on graph $G$, with transitions $W$.*

This result allows us to directly relate the privacy loss from $u$ to $v$ to the probability that the random walk on $G$ with transition probabilities given by the gossip matrix $W$ goes from $u$ to $v$ in a certain number of steps. It thus captures a notion of distance between nodes in the graph. We also report the utility under fixed mean privacy loss $\bar{\varepsilon}=\max_{v\in\mathcal{V}}\bar{\varepsilon}_v\leqslant\varepsilon$ in Table 1 for various graphs, where one can see a utility-privacy trade-off improvement of $n\sqrt{\lambda_W}/d$, where $d$ is the maximum degree, compared to LDP. Using expanders closes the gap with a trusted aggregator (i.e., central DP) up to constant and logarithmic terms. Remarkably, graph topologies that make gossip averaging efficient (i.e. with big $\sqrt{\lambda_W}/d$), such as exponential graphs or hypercubes [59], are also the ones that achieve optimal privacy amplification (up to logarithmic factors). In other words, *privacy, utility and scalability are compatible*.

### 3.3 Private Randomized *Muffliato*

Synchronous protocols require global coordination between nodes, which can be costly or even impossible in some settings. On the contrary, asynchronous protocols only require separated activation of edges: they are thus are more resilient to stragglers nodes and faster in practice. In asynchronous gossip, at a given time-step a single edge $\{u,v\}$ is activated independently from the past with probability

$p_{\{u,v\}}$, as described by Boyd et al. [11]. In our setting, randomized *Muffliato* (Algorithm 2) corresponds to instantiating our general analysis with $W^t = W_{\{v_t,w_t\}} = I_n - (e_{v_t} - e_{w_t})(e_{v_t} - e_{w_t})^\top/2$ if $\{v_t, w_t\}$ is sampled at time $t$. The utility analysis is similar to the synchronous case.

**Theorem 3** (Utility analysis). *Let $\lambda(p)$ be the spectral gap of graph $G$ with weights $(p_{\{v,w\}})_{\{v,w\}\in\mathcal{E}}$. Randomized* Muffliato *(Algorithm 2) verifies, for all $t \geqslant T^{\mathrm{stop}}$:*

$$\frac{1}{2n}\sum_{v\in\mathcal{V}}\mathbb{E}\left[\|x_v^t - \bar{x}\|^2\right] \leqslant \frac{2\sigma^2}{n}, \quad where\ T^{\mathrm{stop}} = \frac{1}{\lambda(p)}\ln\left(\frac{n}{\sigma^2}\max\left(\sigma^2, \frac{1}{n}\sum_{v\in\mathcal{V}}\|x_v^0 - \bar{x}\|^2\right)\right).$$

To compare with synchronous gossip (Algorithm 1), we note that activation probabilities can be derived from a gossip matrix $W$ by taking $p_{\{u,v\}} = 2W_{\{u,v\}}/n$ implying that $\lambda(p) = 2\lambda_W/n$, thus requiring $n$ times more iterations to reach the same utility as the synchronous applications of matrix $W$. However, for a given time-horizon $T$ and node $v$, the number of communications $v$ can be bounded with high probability by a $T/n$ multiplied by a constant whereas Algorithm 1 requires $d_v T$ communications. Consequently, as reported in Table 1, for a fixed privacy mean $\bar{\bar{\varepsilon}}_v$, Algorithm 2 has the same utility as Algorithm 1, up to two differences: the degree factor $d_v$ is removed, while $\sqrt{\lambda_W}$ degrades to $\lambda_W$ as we do not accelerate randomized gossip (see Remark 1 below). Randomized gossip can thus achieve an optimal privacy-utility trade-off with large-degree graphs, as long as the spectral gap is small enough.

**Remark 1** (Accelerating Randomized *Muffliato*). *For simplicity, Randomized* Muffliato *(Algorithm 2) is not accelerated, while* Muffliato *(Algorithm 1) uses Chebychev acceleration to obtain a dependency on $\sqrt{\lambda_W}$ rather than $\lambda_W$. Thus, and as illustrated by Table 1, Algorithm 2 does not improve over Algorithm 1 for all values of $d$ (maximum degree), $n$ and $\lambda_W$. However, Algorithm 2 can be accelerated using a continuized version of Nesterov acceleration [24, 25], thus replacing $\lambda(p)$ in the expression of $T^{\mathrm{stop}}$ in Theorem 3, by $\sqrt{\lambda(p)/(dn)}$. Doing so, using randomized communications improve privacy guarantees over Algorithm 1 for all graphs considered in Table 1.*

### 3.4 Erdös-Rényi Graphs

So far the graph was considered to be public and the amplification only relied on the secrecy of the messages. In practice, the graph may be sampled randomly and the nodes need only to know their direct neighbors. We show that we can leverage this through the weak convexity of Rényi DP to amplify privacy between non-neighboring nodes. We focus on Erdös-Rényi graphs, which can be built without central coordination by picking each edge independently with the same probability $q$. For $q = c\ln(n)/n$ where $c > 1$, Erdös-Rényi graphs are good expanders with node degrees $d_v = \mathcal{O}(\log n)$ and $\lambda_W$ concentrating around 1 [33]. We obtain the following privacy guarantees.

**Theorem 4** (*Muffliato* on a random Erdös-Rényi graph). *Let $\alpha > 1$, $T \geqslant 0$, $\sigma^2 \geqslant \frac{\Delta^2 \alpha(\alpha-1)}{2}$ and $q = c\frac{\ln(n)}{n}$ for $c > 1$. Let $u, v \in \mathcal{V}$ be distinct nodes. After running Algorithm 1 with these parameters, node $u$ is $(\alpha, \varepsilon_{u\to v}^T(\alpha))$-PNDP with respect to $v$, with:*

$$\varepsilon_{u\to v}^T(\alpha) \leqslant \begin{cases} \dfrac{\alpha\Delta^2}{2\sigma^2} & \text{with probability } q\,, \\ \dfrac{\alpha\Delta^2}{\sigma^2}\dfrac{Td_v}{n-d_v} & \text{with probability } 1-q\,. \end{cases}$$

This results shows that with probability $q$, $u$ and $v$ are neighbors and there is no amplification compared to LDP. The rest of the time, with probability $1 - q$, the privacy matches that of a trusted aggregator up to a degree factor $d_v = \mathcal{O}(\log n)$ and $T = \tilde{\mathcal{O}}(1/\sqrt{\lambda_W}) = \tilde{\mathcal{O}}(1)$ [33]. In particular, if several rounds of gossip averaging are needed, as in the next section for SGD, changing the graph mitigates the privacy loss of the rounds where two nodes are neighbors thanks to the rounds where they are not.

## 4 Private Decentralized Optimization

We now build upon *Muffliato* to design decentralized optimization algorithms. Each node $v \in \mathcal{V}$ possesses a data-dependent function $\phi_v : \mathbb{R}^D \to \mathbb{R}$ and we wish to *privately* minimize the function

$$\phi(\theta) = \frac{1}{n}\sum_{v\in\mathcal{V}}\phi_v(\theta), \quad \text{with } \phi_v(\theta) = \frac{1}{|\mathcal{D}_v|}\sum_{x_v\in\mathcal{D}_v}\ell_v(\theta, x_v), \quad \theta \in \mathbb{R}^D, \tag{8}$$

**Algorithm 3:** MUFFLIATO-SGD and MUFFLIATO-GD

---

**Input:** initial points $\theta_v^0 \in \mathbb{R}^D$, number of iterations $T$, step sizes $\nu > 0$, noise variance $\sigma^2$,
gossip matrices $(W_t)_{t \geqslant 0}$, local functions $\phi_v$, number of communication rounds $K$

**for** $t = 0$ *to* $T - 1$ **do**
    **for** *all nodes $v$ in parallel* **do**
        Compute $\hat{\theta}_v^t = \theta_v^t - \nu \nabla_\theta \ell_v(\theta_v^t, x_v^t)$ where $x_v^t \sim \mathcal{L}_v$
    $\theta_v^{t+1} = \text{MUFFLIATO}\big((\hat{\theta}_v^t)_{v \in \mathcal{V}}, W_t, K, \nu^2 \sigma^2\big)$

---

where $\mathcal{D}_v$ is the (finite) dataset corresponding to user $v$ for data lying in a space $\mathcal{X}_v$, and $\ell_v : \mathbb{R}^D \times \mathcal{X}_v \to \mathbb{R}$ a loss function. We assume that $\phi$ is $\mu$-strongly convex, and each $\phi_v$ is $L$-smooth, and denote $\kappa = L/\mu$. We note that our results can be extended to the general convex and smooth setting. Denoting by $\theta^\star$ the minimizer of $\phi$, for some non-negative $(\zeta_v^2)_{v \in \mathcal{V}}$, $(\rho_v^2)_{v \in \mathcal{V}}$ and all $v \in \mathcal{V}$, we assume:

$$\|\nabla \phi_v(\theta^\star) - \nabla \phi(\theta^\star)\|^2 \leqslant \zeta_v^2 \quad , \quad \mathbb{E}\left[\|\nabla \ell_v(\theta^\star, x_v) - \nabla \phi(\theta^\star)\|^2\right] \leqslant \rho_v^2, \quad x_v \sim \mathcal{L}_v,$$

where $\mathcal{L}_v$ is the uniform distribution over $\mathcal{D}_v$. We write $\bar{\rho}^2 = \frac{1}{n} \sum_{v \in \mathcal{V}} \rho_v^2$ and $\bar{\zeta}^2 = \frac{1}{n} \sum_{v \in \mathcal{V}} \zeta_v^2$.

We introduce Algorithm 3, a private version of the classical decentralized SGD algorithm studied in [39]. Inspired by the optimal algorithm MSDA of Scaman et al. [54] that alternates between $K$ Chebychev-accelerated gossip communications and expensive dual gradient computations, our Algorithm 3 alternates between $K$ Chebychev-accelerated gossip communications and cheap local stochastic gradient steps. This alternation reduces the total number of gradients leaked, a crucial point for achieving good privacy. Note that in Algorithm 3, each communication round uses a potentially different gossip matrix $W_t$. In the results stated below, we fix $W_t = W$ for all $t$ and defer the more general case to Appendix F, where different independent Erdös-Rényi graphs with same parameters are used at each communication round.

**Remark 2.** *Our setting encompasses both GD and SGD. Muffliato-GD is obtained by removing the stochasticity,* i.e., *setting $\ell_v(\cdot) = \phi_v(\cdot)$. In that case, $\bar{\rho}^2 = 0$.*

**Theorem 5** (Utility analysis of Algorithm 3)**.** *For suitable step-size parameters, for a total number of $T^{\text{stop}}$ computations and $T^{\text{stop}} K$ communications, with:*

$$T^{\text{stop}} = \tilde{\mathcal{O}}(\kappa), \quad \text{and} \quad K = \left\lceil \sqrt{\lambda_W}^{-1} \ln\left(\max\left(n, \frac{\bar{\zeta}^2}{D\sigma^2 + \bar{\rho}^2}\right)\right) \right\rceil,$$

*the iterates $(\theta^t)_{t \geqslant 0}$ generated by Algorithm 3 verify $\mathbb{E}\left[\phi(\tilde{\theta}^{\text{out}}) - \phi(\theta^\star)\right] = \tilde{\mathcal{O}}(\frac{D\sigma^2 + \bar{\rho}^2}{\mu n T^{\text{stop}}})$ where $\tilde{\theta}^{\text{out}} \in \mathbb{R}^D$ is a weighted average of the $\bar{\theta}^t = \frac{1}{n} \sum_{v \in \mathcal{V}} \theta_v^t$ until $T^{\text{stop}}$.*

For the following privacy analysis, we need a bound on the sensitivity of gradients with respect to the data. To this end, we assume that for all $v$ and $x_v$, $\ell_v(\cdot, x_v)$ is $\Delta_\phi/2$ Lipschitz[2].

**Theorem 6** (Privacy analysis of Algorithm 3)**.** *Let $u$ and $v$ be two distinct nodes in $\mathcal{V}$. After $T$ iterations of Algorithm 3 with $K \geqslant 1$, node $u$ is $(\alpha, \varepsilon_{u \to v}^T(\alpha))$-PNDP with respect to $v$, with:*

$$\varepsilon_{u \to v}^T(\alpha) \leqslant \frac{T \Delta_\phi^2 \alpha}{2\sigma^2} \sum_{k=0}^{K-1} \sum_{w:\{v,w\} \in \mathcal{E}} \frac{(W^k)_{u,w}^2}{\|(W^k)_w\|^2}. \tag{9}$$

*Thus, for any $\varepsilon > 0$, Algorithm 3 with $T^{\text{stop}}(\kappa, \sigma^2, n)$ steps and for $K$ as in Theorem 5, there exists $f$ such that the algorithm is $(\alpha, f)$-pairwise network DP, with:*

$$\forall v \in \mathcal{V}, \quad \bar{\varepsilon}_v \leqslant \varepsilon \quad \text{and} \quad \mathbb{E}\left[\phi(\tilde{\theta}^{\text{out}}) - \phi(\theta^\star)\right] \leqslant \tilde{\mathcal{O}}\left(\frac{\alpha D \Delta_\phi^2 d}{\mu n^2 \varepsilon \sqrt{\lambda_W}} + \frac{\bar{\rho}^2}{nL}\right),$$

*where $d = \max_{v \in \mathcal{V}} d_v$.*

---

[2]This assumption can be replaced by the more general Assumption 2 given in Appendix F.

The term $\frac{\bar{\rho}^2}{nL}$ above (which is equal to zero for *Muffliato*-GD, see Remark 2) is privacy independent. It is typically dominated by the first term, which corresponds to the utility loss due to privacy. Comparing Theorem 6 with the results for *Muffliato* (Table 1 in Section 3.2), the only difference lies in the factor $D\Delta_\phi^2/\mu$. Note that $\Delta_\phi^2$ plays the role of the sensitivity $\Delta^2$, and $D$ appears naturally due to considering $D$-dimensional parameters. On the other hand, $\mu$ is directly related to the complexity of the optimization problem through the condition number $\kappa$: the easier the problem, the better the privacy-utility trade-off of our algorithm. Regarding the influence of the graph, the same discussion as after Corollary 1 applies here. In particular, for expander graphs like the exponential graph of [59], the factor $d/\sqrt{\lambda_W}$ is constant. In this case, converting to standard $(\epsilon, \delta)$-DP gives $\tilde{\mathcal{O}}\big(\frac{D\Delta_\phi^2}{\mu n^2 \epsilon^2}\big)$, recovering the optimal privacy-utility trade-off of central DP [4, 56] up to logarithmic factors. Remarkably, we achieve this optimal rate under a near-linear gradient complexity of $T^{\mathrm{stop}}(\kappa, \sigma^2, n)n = \tilde{\mathcal{O}}(\kappa n)$ and near-linear total number of messages $T^{\mathrm{stop}}(\kappa, \sigma^2, n)Kn = \tilde{\mathcal{O}}(\kappa n)$.

**Remark 3** (Time-varying graphs). *The analysis of* Muffliato-*GD/SGD presented in this section (Theorems 5 and 6) assumes constant gossip matrices $W_t = W$. A more general version of these results is presented in Appendix F to handle time-varying matrices and graphs. This can be used to model randomized communications (as previously described for gossip averaging in Section 3.3) as well as user dropout (see experiments in Appendix H). Time-varying graphs can also be used to split the privacy loss more uniformly across the different nodes by avoiding that nodes have the same neighbors across multiple gossip computations. We illustrate this experimentally for decentralized optimization in Section 5, where we randomize the graph after each gradient step of* Muffliato-*GD.*

## 5 Experiments

In this section, we show that pairwise network DP provides significant privacy gains in practice even for moderate size graphs. We use synthetic graphs and real-world graphs for gossip averaging. For decentralized optimization, we solve a logistic regression problem on real-world data with time-varying Erdos-Renyi graphs, showing in each case clear gains in privacy compared to LDP. The code used to obtain these results is available at https://github.com/totilas/muffliato.

**Averaging on synthetic graphs.** We generate synthetic graphs with $n = 2048$ nodes and define the corresponding gossip matrix according to the Hamilton scheme. Note that the privacy guarantees of *Muffliato* are deterministic for a fixed $W$, and defined by Equation 4. For each graph, we run *Muffliato* for the theoretical number of steps required for convergence, and report in Figure 1(a) the pairwise privacy guarantees aggregated by shortest path lengths between nodes, along with the LDP baseline for comparison. *Exponential graph* (generalized hypercube): this has shown to be an efficient topology for decentralized learning [59]. Consistently with our theoretical result, privacy is significantly amplified. The shortest path completely defines the privacy loss, so there is no variance. *Erdos-Renyi graph* with $q = c\log n/n$ $(c \geqslant 1)$ [22], averaged over 5 runs: this has nearly the same utility-privacy trade-off as the exponential graph but with significant variance, which motivates the time-evolving version mentioned in Remark 3. *Grid:* given its larger mixing time, it is less desirable than the two previous graphs, emphasizing the need for careful design of the communication graph. *Geometric random graph:* two nodes are connected if and only if their distance is below a given threshold, which models for instance Bluetooth communications (effective only in a certain radius). We sample nodes uniformly at random in the square unit and choose a radius ensuring full connectivity. While the shortest path is a noisy approximation of the privacy loss, the Euclidean distance is a very good estimator as shown in Appendix H.

**Averaging on real-world graphs.** We consider the graphs of the Facebook ego dataset [41], where nodes are the friends of a given user (this central user is not present is the graph) and edges encode the friendship relation between these nodes. Ego graphs typically induce several clusters corresponding to distinct communities: same high school, same university, same hobbies... For each graph, we extract the giant connected component, choose a user at random and report its privacy loss with respect to other nodes. The privacy loss given by LDP is only relevant within the cluster of direct neighbors: privacy guarantees with respect to users in other communities are significantly better, as seen in Figure 1(b). We observe this consistently across other ego graphs (see Appendix H). This is in line with one of our initial motivation: our pairwise guarantees are well suited to situations where nodes want stronger privacy with respect to distant nodes.

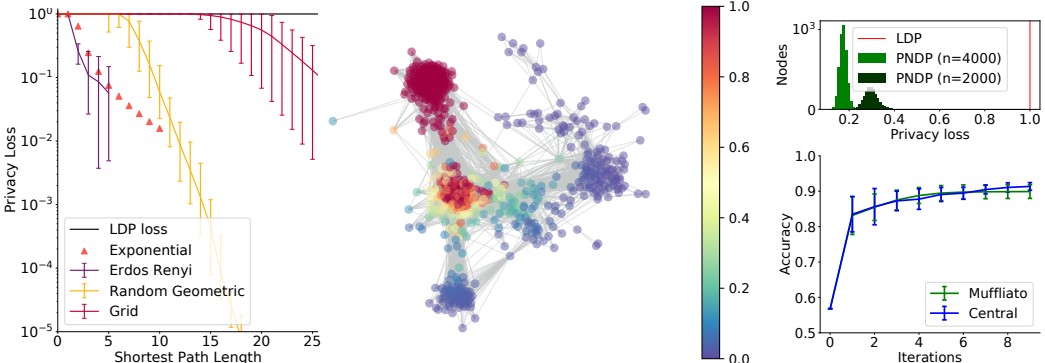

Figure 1: (a) Left: Privacy loss of *Muffliato* in pairwise NDP on synthetic graphs (best, worst and average in error bars over nodes at a given distance), confirming a significant privacy amplification as the distance increases. (b) Middle: Privacy loss of *Muffliato* from a node chosen at random on a Facebook ego graph, showing that leakage is very limited outside the node's own community. (c) Right: Privacy loss and utility of *Muffliato*-GD when using different Erdős-Rényi graphs after each gradient step, compared to a baseline based on a trusted aggregator.

**Logistic regression on real-world data.** Logistic regression corresponds to minimizing Equation 8 with loss function $\ell(\theta; x, y) = \ln(1 + \exp(-y\theta^\top x))$ where $x \in \mathbb{R}^d$ and $y \in \{-1, 1\}$. We use a binarized version of UCI Housing dataset.[3] We standardize the features and normalize each data point $x$ to have unit $L_2$ norm so that the logistic loss is 1-Lipschitz for any $(x, y)$. We split the dataset uniformly at random into a training set (80%) and a test set and further split the training set across users. After each gradient step of *Muffliato*-GD, we draw at random an Erdős-Rényi graph of same parameter $q$ to perform the gossiping step and run the theoretical number of steps required for convergence. For each node, we keep track of the privacy loss towards the first node (note that all nodes play the same role). We report the pairwise privacy loss for this node with respect to all others for $n = 2000$ and $n = 4000$ in Figure 1(c) (top). We see that, as discussed in Remark 3, time-varying graphs are effective at splitting the privacy loss more uniformly across nodes: the privacy gains over LDP are clear with respect to all nodes. As captured by our theory, these gains increase with the number of nodes $n$ in the system, and they also concentrate better around the mean. We compare the utility of *Muffliato*-GD to a federated learning alternative which uses the same parameters but aggregates noisy model updates using a *trusted* central server rather than by gossiping. As seen in Figure 1(c) (bottom), both approaches behave similarly in terms of accuracy across iterations.

## 6 Conclusion

We showed that gossip protocols amplify the LDP guarantees provided by local noise injection as values propagate in the graph. Despite the redundancy of gossip that, at first sight, could be seen as an obstacle to privacy, the privacy amplification turns out to be significant: it can nearly match the optimal privacy-utility trade-off of the trusted curator. From the fundamental building block — noise injection followed by gossip — that we analyzed under the name *Muffliato*, one can easily extend the analysis to other decentralized algorithms, such as the dual approach proposed in [54]. Our results are motivated by the typical relation between proximity in the communication graph and lower privacy expectations. Other promising directions are to assume that closer people are more similar, which leads to smaller individual privacy accounting [27], to design new notions of similarity between nodes in graphs as done in personalization [26] that match the privacy loss variations, and to study privacy attacks [52].

**Acknowledgments** This work was supported by grants ANR-20-CE23-0015 (Project PRIDE) ANR-20-THIA-0014 program "AI_PhD@Lille", ANR-19-P3IA-0001(PRAIRIE Institute), and from the MSR-INRIA joint centre. We thank the organizers of NeurIPS@Paris 2021 for the in-person event that allowed us to meet by chance and to start this piece of work.

---

[3]https://www.openml.org/d/823

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
