# OpenReview forum: "Muffliato: Peer-to-Peer Privacy Amplification for Decentralized Optimization and Averaging"
_NeurIPS.cc/2022/Conference — NeurIPS 2022 Accept_

### Official Review · Reviewer_BroW · 2022-07-11

**Rating:** 7
**Confidence:** 1
**Soundness:** 4 excellent
**Presentation:** 2 fair
**Contribution:** 3 good

**Summary:**

This paper introduces Pairwise network DP, a relaxation of local DP that allows the privacy constraint to vary as a function of nodes in the decentralized graph (i.e. may allow close nodes to lose more privacy than distant nodes). Given this setting, the work introduces Muffliato which combines local noise injection and gossip protocols and it further derives DP optimization algorithms. The theoretical analysis demonstrates the magnitude of privacy amplification for the protocols.  Intuitively, since nodes only interact directly with their neighbors, the this formulation allows for much tighter privacy guarantees for distant nodes and network topology can have a large impact on privacy utility tradeoffs. Finally, they also demonstrate privacy gain on synthetic and graph datasets.

**Questions:**

Suggestion: Additional federated learning experiments on a real world dataset (with a real world graph) demonstrating the practical utility of this method would be helpful.

**Limitations:**

The paper adequately notes its limitations and discusses its social impacts.

**Strengths And Weaknesses:**

To the best of my knowledge, this work offers novel analysis to quantify the privacy amplification of the decentralized protocol for arbitrary graph types and demonstrates a strong improvement in privacy utility tradeoffs over LDP, both theoretically and empirically.

---

> ### Author Response · Authors · 2022-08-02
> **Response to Reviewer BroW**
>
> We thank the reviewer for stressing the novelty of our analysis and the fact that it leads to "strong improvement in privacy utility tradeoffs over LDP" and allows "much tighter privacy guarantees".
>
> Regarding the experiments, we stress that we already use real world datasets, both in term of graphs (Facebook Ego dataset) and with the (admittedly toyish) Housing dataset for gradient descent. These experiments are mostly meant to illustrate our theoretical results: we do provide theoretical convergence guarantees in the paper and run all experiments until the theoretical values of $T^{stop}$. Then, by design, our approach will provide the same improvement between LDP and our relaxation for other learning tasks. Our experiments also illustrate the improvements as the number of nodes increases, and the natural correlations between communities in the graph and the privacy losses. We also added new experiments with the time-varying graphs modeling user dropouts as requested by Reviewer MF8v. Finally, we will release the code on GitHub once published, allowing anyone to experiment further.

---

> > ### Comment · Reviewer_BroW · 2022-08-05
> > **Response to author**
> >
> > Thank you for addressing my concern, I've raised my score.

---

### Official Review · Reviewer_Zauo · 2022-07-12

**Rating:** 7
**Confidence:** 3
**Soundness:** 2 fair
**Presentation:** 3 good
**Contribution:** 2 fair

**Summary:**

The paper studies a relaxation of local model of privacy, which they call pairwise network differential privacy. They use gossip protocols to understand the privacy loss under this relaxed notion. They show that the privacy get amplified under this notion of privacy for a range of network topology.

**Questions:**

Please take a look at above.

**Limitations:**

Yes.

**Strengths And Weaknesses:**

The main strength of the paper is the new notion of privacy studied. It seems natural for certain settings, but I am not sure if it is . They also study their new notion of privacy under the setting of colluding nodes in the network. The ring graph and complete graph was studied recently, so that result is not new. The result that is new are for expanded graphs and random graph. The proof seems correct.

The main weakness of the paper is also the new notion of privacy. For a privacy notion to be of use, it has to satisfy basic properties: group privacy, composition, and robustness to post-processing. I believe the authors can extend their collusion result to get group privacy. I would suggest the authors to do that in order to make this new relaxation more rigorous. It is also unclear what would be a good notion of composition for this new notion of privacy. Do we assume that the network topology (and collusion party) do not change over the period of composition or should we imagine the composition model to be more fitted with their time-varying graph model.

The other weakness of the paper is the interpretability of the result. Theorem 1 is stated with respect to the norm of the gossip matrix (I am guessing it is the spectral norm). It would be nice if we can have a sense of this theorem with an example.

The mean privacy loss as the baseline to compare privacy does not make sense.

I would expect Algorithm 2 should have improvement for all values of d and spectral gap. This does not seem to be the case for arbitrary graphs.

---

> ### Author Response · Authors · 2022-08-02
> **Response to Reviewer Zauo**
>
> We thank the reviewer for his/her feedback and address his/her concerns below.
>
> **On the novelty with respect to [15]**
> While [15] indeed derives guarantees in the case of the ring and the complete graph, we stress that the studied algorithm is not the same: they study random walks where a single token walks along the edges, whereas our analysis holds for gossip and randomized gossip algorithms, which enable parallel computation and thus better scalability. Hence, even for the complete graph, the derivation of the bound is new, as we explain in the related work, line 96-99.
>
>
> **On completing our definition with group privacy, composition and post-processing**
> We thank the reviewer for raising this interesting point, although we believe that this is not always needed for having a meaningful relaxation, for instance we are not aware of specific composition/group privacy/post-processing results for the shuffle model or privacy amplification by decentralization: they often follow from general differential privacy properties. Specifically:
> - For composition, the privacy losses from a node $u$ to a node $v$ in each computation simply sum up (as in classic Rényi DP), and this holds even if the graph is different in each computation. This is what we use to analyze our gradient descent algorithms across the different gradient steps.
> - Post-processing directly follows from the post-processing property of Rényi DP.
> - Group privacy generally relies on the worst database pair at distance $k$ for the adjacency relation considered, see for instance Proposition 2 in the paper introducing Rényi DP [43]. In our setting, this arbitrary choice is not well adapted to the structure of the graph (where we could imagine that a group corresponds to users closely related in the graph) and the fact that privacy losses are different across for each pair of nodes. We agree with the reviewer that we could alternatively analyze group privacy similarly to how we study collusion. We will write it down properly in the final version.
>
> **On the interpretation of Theorem 1**
> Theorem 1 involves the **Euclidean norm of columns** of the gossip matrix, not the spectral norm of the matrix. We will make this more clear.
> We can have a nice interpretation of the result for constant gossip matrices. In this case, the upper bound (righthandside of Equation 6) can be thought of as a distance between nodes $u$ and $v$ in the graph, as explained in Corollary 1 (proved in Appendix C.2). More precisely, $f(u,v)$ in Theorem 1 can be expressed using the probability to be in $v$ when performing a random walk from $u$, which is intuitively decreasing as the distance between $u$ and $v$ increase. The intuition is thus that nodes that are at some non-negligible distance from each other have stronger pairwise privacy guarantees.
>
> **Mean privacy loss as comparison metric**
> In lines 174 and below, we acknowledge the limitations of the mean privacy loss and stress that it should not be taken as a privacy guarantee by itself. However, our pairwise notion of privacy keeps track of privacy losses for each pair of nodes, giving us $\mathcal{O}(n^2)$ numbers compared to a single one for central DP. The mean privacy loss allows to make a comparison and has two advantages:
> - it can be interpreted as the level of "attack" that a node can perform on all the other nodes of the graph. If the target is chosen randomly, then the expected privacy loss is the mean privacy loss.
> - it is computationally tractable and gives us closed-form formulas that are easy to compare.
>
> We emphasize that all our results do not only bound the mean privacy loss but also provide formulas for each pairwise privacy loss that can be evaluated numerically, but are less interpretable (*cf* the previous point on the interpretation of Theorem 1).
>
> **Improvements of Algorithm 2 for arbitrary graphs**
> Indeed, Algorithm 2 does not improve upon Algorithm 1 all values of $d$ and spectral gap, as highlighted in Table 1. However, and as we explain in Footnote 2 (page 7), this is due to the fact that our randomized Muffliato algorithm is not accelerated, while Algorithm 1 uses Chebychev acceleration. Algorithm 2 can be accelerated by using the continuized version of Nesterov acceleration [23]. Doing so, Algorithm 2 would provide improvements over Algorithm 1 for all values of $d$ and spectral gap.
> We did not pursue accelerating Algorithm 2 in an effort to keep the paper easy to follow, due to the technicality of introducing the acceleration presented in [23]. We will expand Footnote 2 into a remark where we clarify this.
>
> We hope that we addressed the main reviewer's concerns, and that our explanations and the improvements we described will convince him/her to increase his/her score.

---

> > ### Comment · Reviewer_Zauo · 2022-08-06
> > **Response to response**
> >
> > I really appreciate the authors for their response. I think they have answered my questions.
> >
> > I believe the reason for no group privacy result for the shuffle model also follows from a lack of proper adversarial definition. It has been true even in cryptography from where the shuffle model is borrowed (IKOS paper).
> >
> > For composition, I might be wrong, but I think the authors are considering non-adaptive composition and settings where the underlying graph topology remains static.
> >
> > I am increasing the score because I really think that the privacy notion is interesting.
> >
> > For my own understanding (the authors can respond even after the entire reviewing process): is there something fundamentally different or intuition to be achieved by using gossip protocols to understand the privacy on rings and complete graph?

---

### Official Review · Reviewer_MF8v · 2022-07-18

**Rating:** 7
**Confidence:** 3
**Soundness:** 4 excellent
**Presentation:** 4 excellent
**Contribution:** 3 good

**Summary:**

The authors propose a relaxed notion of local differential privacy in a
distributed model of local optimization in which users share data across
a network over rounds. The key difference from LDP is that for a particular
round, a user only sees data from his neighbors, and thus data from many hops
away is aggregated more and is more private. They propose muffliato, an algorithm
for computing noisy averages with an improved accuracy guarantee over LDP for
several important types of network topologies. They show how to use
muffliato to perform SGD privately.


**Questions:**

Theorems 2,3 hold for a particular value of T^{stop} which is upper bounded as
stated and not for all values of T^{stop} up to that bound, right? The second
case seems incorrect as early iterates would not be accurate. Please make this
more clear in the theorem statements.

In real life, the communication networks often change between rounds of
interaction, and users drop out and come back online. This seems to be
captured well by the general analysis of
Muffliato, which allows W to change over time. However, for random muffliato,
how realistic is it to assume that some mechanism exists to sample edges randomly
and query the users? Does this require a central authority to do the sampling,
and what happens if the users are offline?

This extends to the Muffliatio-GD, where it appears one must assume a fixed
communication matrix, or use randomized muffliato, many times.


**Limitations:**

The communication/coordination limitation could be addressed by running experiments
with different communication graphs each round.

(Edit) I thank the authors for addressing this concern.

**Strengths And Weaknesses:**

+ muffliato offers a big improvement over LDP in terms of accuracy and realistic
assumptions about the network in which the compuatations are done.

+ The authors provide a tight privacy analysis of Muffliato when the Gaussian
  mechanism is used.

+ the experiments are good quality and indicate the improvements of Muffliato

- Muffliato, and especially Muffliato-SGD, still require many rounds of
  coordination between users which is often hard to achieve in the decentralized
  setting.

---

> ### Author Response · Authors · 2022-08-02
> **Response to Reviewer MF8v**
>
> We thank the reviewer for the positive feedback and address his/her questions below.
>
> **Clarification in theorems 2 & 3**
> Theorems 2 and 3 are convergence results, they indeed hold for all time steps greater (and not lower) than the particular value of $T^{\rm stop}$ given by the theorem. We will correct our formulation.
>
> **On realistic implementations in building graphs/communications**
> Gossiping with a constant gossip matrix $W$ and performing synchronous decentralized algorithms with such a constant matrix (such as e.g., [7,48]) indeed requires costly synchronization. However, this problem is typically addressed using randomized communications (randomized gossip matrices, as in Algorithm 2, or [10,23]): they do not require any global coordination. Indeed, randomized communications can be generated through the use of local Poisson point processes at each node, as explained in references [10,23].
>
>
> **Offline / unavailable users**
> As stressed wisely by the reviewer, user dropouts can be dealt with using time-varying matrices, which is well captured by the general analysis of Muffliato (Theorem 1). In randomized Muffliato, users that may be offline may also be modeled by very small probability of activations (eventually equal to 0) when they are offline. Then, using the generality of our analysis (in Theorem 1 e.g.), time-varying probability activations can be handled in the same way as time varying gossip matrices. Hence, our results already include tools to handled correctly dropout, which of course can be modeled in different ways.
> Even though our analysis of Muffliato-GD in the main text uses a constant communication matrix $W$ in an effort to ease notations, time-varying communication matrices $(W_t)$ are used in the analysis in Appendix F for Muffliato GD. However, the gossip matrix needs to remain constant between each gradient step for  the utility analysis we perform (in order to have a Chebychev acceleration). Using time varying matrices for every communications with acceleration could be done using results of [37], rather than a more classical Chebychev acceleration, at the cost of a more complex algorithm. In the GD experiments, we do change the sampling of the Erdös-Renyi graph after each gradient descent step, thus only assuming that coordination is possible for the time of a single step, and not the whole algorithm.
> To complement our experiments, we added new plots for decentralized averaging in the supplemental material (see files error_dropout.pdf and privacy_dropout.pdf) where we change the graph at each iteration and we model dropouts explicitly. Specifically, at each time step, the availability of each node is modeled by an independent Bernouilli random variable and we draw a new Erdos Renyi graph over the set of available nodes. We vary the expected level of available nodes at each step from 10 to 90% and observe that the convergence time increases with the proportion of inactive users, but the achievable privacy-utility trade-off remains approximately the same (the plot shows a single random run, hence the minor variations). We will add an appendix section explaining these new simulations with more extensive parameter exploration and averaging over several random seeds.
>
> We hope that our answers completely lift the reviewer's concerns and that these details will convince him/her to increase his/her score.

---

> > ### Comment · Reviewer_MF8v · 2022-08-04
> > **Addressing my concerns**
> >
> > I thank the authors for clarifying my questions and addressing my concerns, and I have raised my review accordingly.

---

### Official Review · Reviewer_sTzh · 2022-07-26

**Rating:** 7
**Confidence:** 4
**Soundness:** 4 excellent
**Presentation:** 4 excellent
**Contribution:** 3 good

**Summary:**

This paper proposes pairwise network differential privacy (PNDP) in decentralized optimization that relates privacy loss between two nodes with their communication weights after running the proposed algorithms for T steps. Several mixing matrices are considered and the paper analyzes both simple gossip averaging and baseline decentralized optimization problems. Utility and privacy analyses are conducted for different mixing matrices and problems. The new notion of differential privacy averages privacy loss for all nodes and thus gives a new perspective of designing privacy-oriented algorithms for decentralized optimization rather than only looking at worst case privacy loss.

**Questions:**

I read the proof of Synchronous Muffiato and have the following questions. 1. In line 590, how to get the first bias-variance decomposition? It seems to the cross term has dependent parts and cannot cancel. 2. The first inequality here seems to rely on some properties of $P_t(W)$ while dealing with the 2nd term in the 2nd line. 3. The first term in the 2nd line misses an expectation symbol. in the first line, I guess there should be $\|x^t - \mathbb{1} \bar{x}^\top\|$ otherwise the dimension of $\bar{x}$ is confusing. 4. How to reach the inequality in line 495?

**Limitations:**

Yes.

**Strengths And Weaknesses:**

The idea of PNDP is a pairwise version of network DP from ref. [16], which highlights and averaged privacy loss for each node instead of the worst case privacy loss. This perspective is original. This paper has quality analyses and experiments to support its claims and is clearly written. For each case considered, the paper shows clear and solid theoretical guarantees. I think this paper is significant in introducing this new idea of pairwise privacy notion, but the main confusion when I read the paper is that how to explicitly compare the utility loss given the same privacy constraints since there is a conversion of DP and Renyi DP which I didn't figure out how these results are compared in detail.

---

> ### Author Response · Authors · 2022-08-02
> **Response to Reviewer  sTzh**
>
> We thank the reviewer for her/his meticulous review, the time spent reviewing the paper and checking the proofs. It will allow us to correct typos and make our proofs more clear.
>
> **On the comparison with classical $(\epsilon, \delta)$-DP**
> We stated all our results in Rényi DP (RDP) because RDP is increasingly popular in differential privacy papers (due to its nice properties) and appears to be becoming the new standard. Note however that we briefly recap at line 133 the conversion from RDP to classical $(\epsilon, \delta)$-DP, and we can of course compute explicitely the guarantees in $(\epsilon, \delta)$-DP for a given result. For instance, a randomized gossip on the complete graph, being $(\alpha, \epsilon)$-RDP with an utility of $\alpha \Delta^2/ n^2 \epsilon$ means that for a utility of $u = \mathbb{E}(\lVert \bar{x} - x^{out} \rVert^2)$ the algorithm is $(\Delta \sqrt{\ln(1/\delta)} / n \sqrt{u}, \delta)$ for all $\delta >0$, compared to $(\Delta \sqrt{\ln(1/\delta)} / \sqrt{nu}, \delta)$ for the local DP counterpart. Hence, we match the classical amplification of order $\sqrt{n}$ that central DP provides over local DP.
> We can add a remark or an annex with a conversion of Table 1 to $(\epsilon, \delta)$-DP.
>
>
> **On the technical details of proofs**
> We clarify the parts pointed out by the reviewer.
> -  Points 1 & 2: We start the proof at line 590, which writes $\frac{1}{2}\mathbb{E}\|x^t-\bar x\|^2 = \frac{1}{2}\mathbb{E}\|P_t(W)(x^{(0)}-\bar{x})\|^2$. Since we want to use the property that we mention at line 587 of the paper (borrowed from Berthier et al. [7]), and since $\frac{1}{n}\sum_{v\in\mathcal{V}}x^{(0)}_v=\bar x + \bar \eta$ (due to $x^{(0)}_v=x_v+\eta_v$), we write $\frac{1}{2}\mathbb{E}\|P_t(W)(x^{(0)}-\bar{x})\|^2=\frac{1}{2}\mathbb{E}\|P_t(W)(x^{(0)}-\bar{x} -\bar \eta +\bar \eta)\|^2=\frac{1}{2}\mathbb{E}\|P_t(W)(x+\eta-\bar{x} -\bar \eta +\bar \eta)\|^2$. Then, because $x+\eta-\bar x-\bar \eta$ is 0-mean with respect to the summation of the coordinates of these vectors (not with respect to $\mathbb{E}$), $P_t(W)(x+\eta-\bar x-\bar \eta)$ is also 0-mean, and since $P_t(W)\bar \eta=\bar \eta$, we use a bias-variance decomposition (with respect to the summation and the coordinates, not wrt $\mathbb{E}$) to obtain $\frac{1}{2}\mathbb{E}\|x^t-\bar x\|^2=\frac{1}{2}\mathbb{E}\|P_t(W)(x+\eta-\bar{x}-\bar\eta)\|^2+\frac{1}{2}\mathbb{E}\|\bar \eta\|^2$, which answers point 1. Then, we use $\frac{1}{2}\mathbb{E}\|\bar \eta\|^2=\frac{\sigma^2}{2n}$ and the property of $P_t(W)$ to get $\frac{1}{2}\mathbb{E}\|P_t(W)(x+\eta-\bar{x}-\bar\eta)\|^2\leq (1-\sqrt{\lambda_W})^t\mathbb{E}\|x+\eta-\bar{x}-\bar\eta\|^2$, leading to the third line in the proof. To obtain the final result, we use again a bias-variance decomposition, with respect to $\mathbb{E}$, to obtain $\mathbb{E}\|x+\eta-\bar{x}-\bar\eta\|^2=\|x-\bar{x}\|^2+\mathbb{E}\|\eta-\bar\eta\|^2\leq \|x-\bar{x}\|^2+\mathbb{E}\|\eta\|^2=\|x-\bar{x}\|^2+n\sigma^2$, concluding this proof.
> - Point 3: there is indeed an expectation missing, and you are also right about the other typo.
> - Point 4: the inequality in line 595 is obtained using Cauchy-Schwarz inequality as written in the paper, in the following way:
>  $\big( \sum\_{w'}(W^t)\_{ww'}\big)^2\leq \big(\sum\_{w'}(W^t)\_{ww'}^2\big)\big(\sum\_{w'} 1\big)$ which can be rewritten further $ n (\sum_{w'}(W^t)_{ww'}^2)=n \rVert (W^t)_w \lVert ^2$ .
>
> We hope that the above details answer totally the four questions, and thank the reviewer for helping us clarify the proof. We will make sure to include this level of detail in the final version.

---

### Author Response · Authors · 2022-08-02
**Global response**

We thank all the reviewers for their useful feedback and their positive comments. In particular, we are happy that reviewers think that our relaxation offers a "new perspective of designing privacy-oriented algorithms for decentralized optimization" (Reviewer sTzh) and provides "a big improvement over LDP in terms of accuracy" (Reviewer MF8v) and a "strong improvement in privacy utility tradeoffs over LDP" (Reviewer BroW), with "realistic assumptions" (Reviewer MF8v).
We address the concerns and questions raised by the reviewers in separate comments below and will remain available during the discussion period if any point remains unclear.

---

### Meta-Review · Area_Chair_xyff · 2022-08-22

**Recommendation:** Accept
**Confidence:** Certain

**Metareview:**

The paper eventually received a perfectly consistent evaluation from all the reviewers (4 times "accept"), so I can only recommend the acceptance.

**Award:**

No

---

### Decision · Program_Chairs · 2022-09-14

Accept